# Bioactive Materials in Vital Pulp Therapy: Promoting Dental Pulp Repair Through Inflammation Modulation

**DOI:** 10.3390/biom15020258

**Published:** 2025-02-10

**Authors:** Liang Qiao, Xueqing Zheng, Chun Xie, Yaxin Wang, Lu Ye, Jiajia Zhao, Jiarong Liu

**Affiliations:** 1Department of Stomatology, Union Hospital, Tongji Medical College, Huazhong University of Science and Technology, Wuhan 430022, China; m202372641@hust.edu.cn (L.Q.); zhengxq@hust.edu.cn (X.Z.); 2009xh1204@hust.edu.cn (C.X.); m202372650@hust.edu.cn (Y.W.); m202372640@hust.edu.cn (L.Y.); 2School of Stomatology, Tongji Medical College, Huazhong University of Science and Technology, Wuhan 430030, China; 3Hubei Province Key Laboratory of Oral and Maxillofacial Development and Regeneration, Wuhan 430022, China

**Keywords:** bioactive materials, vital pulp therapy, inflammation response, pulp repair, pulp healing, molecular mechanism

## Abstract

With the paradigm shift towards minimally invasive biologic therapies, vital pulp therapy (VPT) has been receiving increasing attention. Currently, bioactive materials (BMs), including MTAs, Biodentine, Bioaggregate, and iRoot BP Plus, are clinically widely used for the repair of damaged pulp tissue. Emerging evidence highlights the crucial role of inflammation in pulp repair, with mild to moderate inflammation serving as a prerequisite for promoting pulp repair. BMs play a pivotal role in regulating the balance between inflammatory response and reparative events for dentine repair. Despite their widespread application as pulp-capping agents, the precise mechanisms underlying the actions of BMs remain poorly understood. A comprehensive literature review was conducted, covering studies on the inflammatory responses induced by BMs published up to December 2023. Sources were identified through searches of PubMed and MEDLINE databases, supplemented by manual review of cross-references from relevant studies. The purpose of this article is to discuss diverse mechanisms by which BMs may regulate the balance between tissue inflammation and repair. A deeper understanding of these regulatory mechanisms will facilitate the optimization of current pulp-capping agents, enabling the development of targeted regenerative strategies to achieve superior clinical outcomes.

## 1. Introduction

Dental pulp is enclosed by hard dentin walls [1,2], and it is very common for it to be infected by tooth decay, trauma, or other factors. Inflammation of dental pulp caused by such factors can result in intense pain due to the non-expandable nature of the pulp cavity [3]. In the past, treatment involved exposure and subsequent extraction of the pulp, commonly referred to as Root Canal Treatment (RCT). However, with a paradigm shift towards minimally invasive biologic therapy, vital pulp therapy (VPT) and regenerative endodontic therapy (RET) are increasingly important [4,5]. VPT entails the removal of damaged or infected parts of the pulp tissue, followed by the application of medication or materials to promote repair, ultimately aiming to restore the tooth’s function and esthetics [6]. The advancement of these treatment approaches is strongly correlated with the development of bioactive dental materials.

The most critical step of VPT is ensuring the repair of the dental pulp. In the past, pulpal inflammation was considered an undesirable response, often resulting in cell necrosis and treatment failure. However, recent evidence suggests that inflammation plays a crucial role in the process of pulp repair, which is a prerequisite for inducing pulp repair and healing [7]. At the onset of injury, pattern recognition receptors, including Toll-like receptors (TLRs) on the surface of dentinogenic cells and pulp fibroblasts, bind to relevant molecules and initiate an inflammatory cascade response. Then, pulp fibroblasts and inflammatory cells secrete numerous pro-inflammatory cytokines, amplifying the inflammatory process [8,9]. In the later stages of inflammation, anti-inflammatory cytokines are secreted to terminate inflammation. Notably, some cytokines exert diametrically opposed pro-inflammatory and anti-inflammatory effects through mediating different signaling pathways. When inflammation is reduced to a low level, the tissue microenvironment changes, and the balance shifts towards restorative repair [10]. Dental pulp stem cells (DPSCs) can differentiate in multiple directions, facilitating pulp regeneration and dentin remineralization [11]. In VPT, choosing an appropriate pulp-capping material is crucial for the modulation of the inflammation and repair course.

Bioactive materials (BMs) are multifunctional composite materials composed of ceramics, metals, or polymers and are capable of interacting with living organisms and producing specific biological effects [12]. Given the absence of a universally accepted standard definition of the term and the necessity of its usage, we categorize compounds or complexes containing metal ions that are discussed herein as BMs. These substances act on dental and periodontal tissues, release their own components, and induce restorative changes in the organism. Over the past few decades, significant progress has been made in the development of dental BMs with the ability to interact with surrounding dental tissues and stimulate the repair of pulpal and periradicular tissues. Dental BMs, including mineral trioxide aggregates (MTAs), Biodentine, Bioaggregate, and iRoot BP Plus, are mainly based on calcium silicates and are widely used clinically to repair and regenerate damaged pulp tissue. Currently, the biological properties of BMs and their interactions with the pulp in VPT are better understood [13]. However, numerous studies indicate that BMs appear to exert different effects on inflammation during various stages of the pulpal inflammatory response [14]. And it seems that inducing inflammation rapidly at the initial stage of pulp damage, controlling the severity of inflammation during the process, and inhibiting inflammation at an appropriate time to avoid adverse effects are key factors in promoting the repair of inflamed pulp [8]. Understanding the mechanisms by which BMs influence pulp inflammation and repair will facilitate the development of drugs targeting both pulp repair and inflammation. This review summarizes various mechanisms through which BMs regulate inflammation and repair during VPT.

## 2. Inclusion and Exclusion Criteria

The inclusion criteria were as follows:Original articles*Human* or animal cell culture studies or animal studies

The exclusion criteria were as follows:Case study reportsReview or systematic reviewCommentaries/letters to the editor/expert opinionNon–English-language articles

### Search Methodology

The MEDLINE/PubMed library databases were queried for relevant articles on the topic of applications of bioactive materials in VPT published up to December 2023 (last accessed 31 December 2023). The search terms were the following keywords used in various combinations: “Mineral trioxide aggregates”, “Biodentine”, “iRoot BP Plus”, “pulp capping”, “lithium”, “zinc”, “Strontium”, “Magnesium”, “Silver”, “inflammation”, “molecular mechanism”, “signaling pathway”, “dental stem cell”, “apical papilla stem cell”, and “dental pulp”.

An initial literature search using different combinations of the search terms yielded 1112 articles (Table 1). Figure 1 presents a flowchart of the review process. Titles and abstracts of these articles were reviewed by 2 independent examiners who excluded nonqualifying publications.

## 3. Quality Assessment

In order to enhance the transparency and reliability of the studies, a brief assessment of the quality of the included studies was conducted, and the evidence was categorized into the following three levels based on the GRADE framework: high quality, moderate quality, and low quality. Concurrently, given the heterogeneity of study designs, particular attention was devoted to the randomization of trials, the sample size, the configuration of control groups, and the objectivity of outcome assessment.

High quality: randomized controlled trials and repeated experiments; subjects are dental pulp stem cells or other stem cells within the pulp chamber.Moderate quality: small sample-size experiments or non-randomized controlled trials; subjects are other cells with stemness.Low quality: uncontrolled trials; subjects are mismatched cell types or cells of unknown origin, or animal experiments, or trials that do not fully meet the criteria for high and moderate quality.

## 4. Bioactive Materials

Calcium hydroxide (Ca(OH)_2_) has traditionally been regarded as the gold standard for VPT. Due to its high basicity, calcium hydroxide can locally damage pulp tissue, creating an uncontrolled necrotic area that triggers a sustained inflammatory response and may lead to intra-pulpal calcifications [15]. Currently, calcium hydroxide is being replaced by a new generation of materials—calcium silicate-based materials, which exhibit excellent biocompatibility, intrinsic osteoconductive activity, and the ability to induce regenerative responses. They can promote the formation of higher-quality dentin bridges and improve the sealing of pulp-capped sites [16]. MTA, the first bioceramic material used in endodontics, was developed based on Portland cement and is composed mainly of calcium, silicon, aluminum, bismuth, and iron. With its superior biocompatibility and sealing properties, MTA has been widely used in a variety of VPT and RET and has become the benchmark for the development of novel bioceramic materials [17,18,19]. Biodentine, a “dentine replacement” material developed by Septodont (Saint-Maur-des-Fossés, France) in 2009, contains tricalcium silicate, calcium carbonate, zirconium oxide, and calcium chloride and is widely recognized as a promising material that exhibits excellent physical and biological properties in VPT and RET, including sealing, dentine formation, and pulp regenerative abilities similar to MTA. And Biodentine has a shorter setting time and is less likely to cause tooth discoloration than MTA [20,21,22]. iRoot BP Plus, developed by Innovative Bio Ceramix lnc. (Vancouver, BC, Canada), is regarded as an alternative to MTA. It is composed of tricalcium silicate, zirconium oxide, tantalum pentoxide, dicalcium hydroxysilicate, calcium sulfate, calcium dihydrogen phosphate, and fillers [23]. The literature shows that iRoot BP Plus has been used in various clinical procedures, such as direct or indirect pulp capping and pulpotomy [24,25,26]. The available studies, though limited in number, demonstrate the excellent physicochemical and biological properties of the substance. However, further research is necessary to determine its efficacy. In addition to commercialized bioceramic materials, BMs can also include bioactive ions like titanium and strontium [27,28], bioactive proteins such as amelogenin and *Human* β-defensin 4 [29,30], and naturally occurring active materials like propolis [31].

## 5. Inflammation in Repair of Pulp–Dentin Complex

Inflammation is a natural defense response to damage to the pulp–dentin complex; it aims to eliminate the initial damage factor and initiate the repair process. However, excessive or persistent inflammation may lead to further damage to the pulpal tissues and disease progression. During the initial stages of pulpal injury, immune cells quickly gather at the affected site. Neutrophils are the first immune cells to arrive, and they eliminate invading microorganisms by releasing enzymes and reactive oxygen species [32]. The inflammatory response is then amplified by the recruitment of macrophages and dendritic cells, which are responsible for the phagocytosis of bacteria, the removal of necrotic tissue, and antigen presentation [8]. The onset, persistence, and resolution of acute inflammation in dental pulp are complex biological processes involving multiple immune cellular and molecular pathways. This primarily includes the following aspects.

### 5.1. Interleukin

Both IL-1α and IL-1β, as members of the interleukin family, are ubiquitously expressed and key pro-inflammatory cytokines. IL-1α precursors are released as biologically active mediators during cell necrosis, and they bind to IL-1 receptor 1 (IL-1R1), inducing the same pro-inflammatory effects as IL-1β [33]. IL-6 is expressed during acute inflammation and is primarily produced by T cells, monocytes, fibroblasts, and macrophages in response to antigens [10,34,35]. It is typically regarded as a pro-inflammatory marker within the first 24 h of dental pulp tissue infection. However, it has also been reported that IL-6 has anti-inflammatory effects [36,37,38]. This may be due to the fact that the effects of IL-6 are related to the microenvironment and the timing of its expression. Therefore, the bidirectional effects of IL-6, both pro-inflammatory and anti-inflammatory, are important for the development of inflammatory processes in dental pulp. IL-6 can not only produce regenerative or anti-inflammatory effects through the classical pathway of Mitogen-Activated Protein Kinase (MAPK) signaling but also mediate the pro-inflammatory effects through the trans signaling pathway. IL-8 is produced by cells expressing TLRs, such as dentinogenic cells, neutrophils, and monocytes, but is only released under inflammatory conditions [39,40]. IL-8 recruits and activates neutrophils, induces superoxide production, and enhances the expression of neutrophil adhesion molecules [41,42]. It is evident that IL-8 plays a role in the development of inflammation in pulpitis. And the successful detection of IL-6 and IL-8 upregulation has been recognized as a marker of the induction of pulpitis in numerous studies.

### 5.2. Complement

As an important part of the immune system, the complement system plays a key role in the initiation of pulpal inflammation, elimination of bacteria or irritants, and repair of the pulp–dentin complex. In pulpal inflammation, activation of the complement system can amplify the inflammatory response and promote lesion progression [43,44,45]. The complement system can be activated via the classical pathway, the mannan-binding element pathway, and the alternative pathway, all of which can be induced by pathogens or components of pathogens, such as bacterial lipopolysaccharides. This activation leads to the production of various complement molecules, including C3a, C5a, and membrane attack complex. C3a and C5a are chemotactic factors that attract leukocytes, such as neutrophils and macrophages, to sites of inflammation. They also increase vascular permeability and promote the infiltration of inflammatory mediators [46,47]. Moreover, it has been demonstrated that C5a stimulates odontogenic differentiation through various pathways in both healthy and inflammatory states of *Human* dental pulp stem cells (hDPSCs) [45,48], while C3a facilitates the mobilization and specific recruitment of DPSCs and dental pulp fibroblasts [49].

### 5.3. Cellular Autophagy

Cellular autophagy is an intracellular degradation pathway that maintains cellular homeostasis by encapsulating and digesting discarded or damaged organelles and proteins inside the cell. This process can be activated in various cell types, including pulp fibroblasts, dentinogenic cells, macrophages, and lymphocytes, in response to stress and hypoxia in the inflamed pulp. Autophagy maintains DPSC homeostasis by degrading intracellularly damaged organelles and biomolecules [50]. It also reduces oxidative stress and inflammatory signaling by inhibiting the activation of the NF-κB pathway and the NOD-like receptor thermal protein domain associated protein 3 (NLRP3) inflammasome [51,52]. Futhermore, it is also able to reduce lipopolysaccharide (LPS)-induced pulp cell pyroptosis [53]. Autophagy can protect odontoblasts during early inflammatory stages of caries [50]. However, excessive autophagy induced by stress conditions can cause and exacerbate tissue damage. In the later stages of inflammation, autophagy contributes to the removal of inflammatory cell debris and promotes tissue repair [54,55,56].

### 5.4. Macrophages

Macrophages can exist in either an M1 or M2 polarization state, depending on the microenvironment. These cells play a dual role in inflammation and tissue repair [57]. During the initiation of pulpitis, macrophages activate the immune response by phagocytosing pathogens and releasing inflammatory mediators such as TNF-α, IL-1β, and IL-6. This pro-inflammatory activity is the primary function of M1-type macrophages. At the same time, M1-type macrophages produce ROS to kill pathogens, but this may also cause damage to the surrounding pulp tissue. During the regression and healing phase of inflammation, macrophages undergo M2-type polarization. In their M2 state, macrophages help clean the site of inflammation by phagocytosing cellular debris and dead cells. They also release anti-inflammatory cytokines, including IL-10 and TGF-β, as well as factors that promote cellular proliferation and tissue repair. This promotes the regression of inflammation and the repair of tissues [7,58,59,60].

### 5.5. Molecular Signaling Pathways (Figure 2)

Signaling pathways are fundamental to life processes, mediating the transmission of extracellular molecular signals across cell membranes to exert their effects intracellularly. Extracellular molecular signals (referred to as ligands) include a wide variety of substances, such as hormones, growth factors, cytokines, neurotransmitters, and small molecule compounds. Signaling pathways related to inflammation, including nuclear factor kappa-B (NF-κB), Wnt, Notch, MAPK, and NLRP3, are activated or inhibited to regulate inflammation directly or indirectly by altering the microenvironment.

The NF-κB pathway is crucial in pulpal inflammation as it controls the expression of various inflammatory factors, including TNF-α and IL-1β, which aggravate the inflammatory response and contribute to further damage to pulpal tissue [61]. TNF-α, produced by activated macrophages and T cells, induces the release of inflammatory mediators, recruits lymphocytes and monocytes, and stimulates endothelial cells to express adhesion molecules such as vascular cell adhesion protein 1 (VCAM-1) and intercellular cell adhesion molecule-1 (ICAM-1), as well as secrete chemokines like C-C motif chemokine ligand 2 (CCL2)/monocyte chemoattractant protein 1 (MCP-1) and IL-8 [62,63]. IL-1β, secreted by macrophages, dendritic cells, and dentin-forming cells in the dental pulp following pathogen recognition or stimulation [30], enhances the recruitment and activation of neutrophils and macrophages, increasing their phagocytic activity. This creates a positive feedback loop, stimulating the production of more pro-inflammatory cytokines. Additionally, IL-1β increases the permeability of the pulpal vasculature, promoting the diffusion of inflammatory mediators and exacerbating the inflammatory response [64,65].

MAPKs are upstream components of NF-κB, comprising three families: ERK, JNK, and p38 MAPK [66]. When stimulated by pathogens or activated by inflammatory mediators such as TNF-α and IL-1β, p38 MAPK activates the NF-κB pathway. This pathway regulates the expression of inflammatory genes, promotes the recruitment and activation of inflammatory cells, and ultimately leads to increased inflammation [67,68]. The process is regulated by positive feedback [69]. Additionally, the MAPK pathway regulates pulp cell survival and apoptosis, which are pivotal in determining the extent of pulp tissue damage and its subsequent repair [9].

**Figure 2 biomolecules-15-00258-f002:**
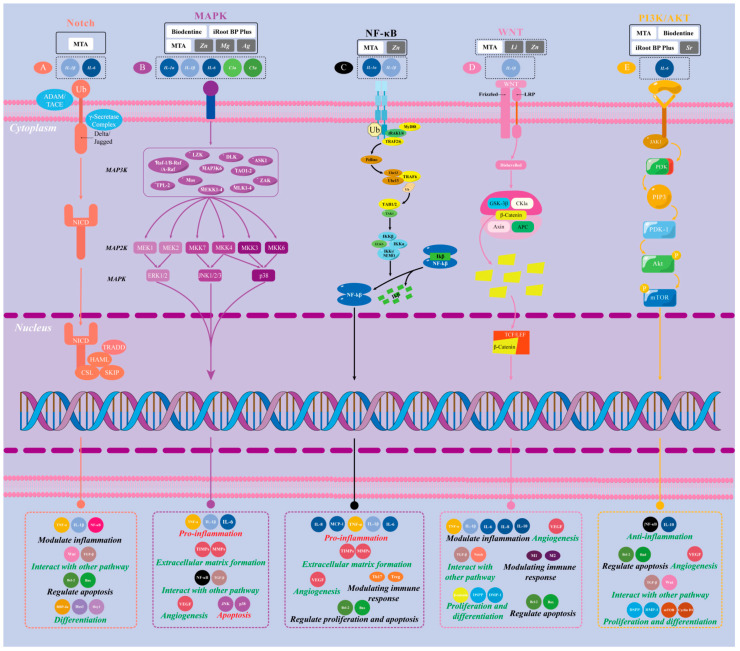
Schematic diagram of the mechanisms of bioactive materials and cytokines in regulating inflammation and maintaining pulp equilibrium through various signaling pathways involving the nucleus [70,71,72,73,74,75,76,77,78]. The illustration shows how signaling pathways control gene expression through transcription factors, affecting inflammation, cell growth, death, and matrix formation in endodontitis. (**A**) The Notch signaling pathway can be activated by MTA, IL-1β, and IL-6. The Notch receptor binds to ligands (e.g., Delta and Jagged) and releases Notch intracellular structural domains (NICDs) in the cytoplasm via ADAM/TACE-mediated ligand shearing. The NICDs then bind to transcription complexes in the nucleus, thereby regulating gene expression. The Notch signaling pathway regulates the inflammatory response, influences multiple pathway interactions, and modulates cell differentiation and apoptosis. (**B**) MAPK signaling pathway can be activated by Biodentine, MTA, iRoot BP Plus, Zn, Mg, Ag, complement C3a, C5a fragments, and various cytokines. Activation of the pathway is initiated by MAP3K, MAP2K, and MAPK family members (e.g., ERK1/2, JNK1/2/3, p38) via progressive signaling by kinases such as leucine-zipper and sterile-α motif kinase (LZK), Apoptosis Signal Regulating Kinase 1 (ASK1), Transforming Growth Factor-β-Activated Kinase 1-v-raf murine sarcoma viral oncogene homolog B (TAK1-B-Raf), and others. The MAPK pathway promotes inflammatory processes, modulates extracellular matrix formation, regulates angiogenesis, and controls apoptosis. (**C**) The NF-κB signaling pathway can be activated by MTA, Zn, IL-1β, and IL-6. Upon activation, IκB is degraded, releasing NF-κB subunits (e.g., p65/p50), which promote the transcription of target genes upon transfer to the nucleus. This pathway exerts robust pro-inflammatory effects, regulates extracellular matrix formation and angiogenesis, mediates cell proliferation and necrosis, and modulates the immune response. (**D**) The WNT signaling pathway can be activated by MTA, Li, Zn, and IL-1β. WNT proteins bind to Frizzled and LRP receptors, inhibiting glycogen synthase kinase-3β (GSK-3β) activity and preventing the degradation of β-catenin, which accumulates and translocates to the nucleus. WNT regulates inflammatory responses, promotes angiogenesis, and modulates cell proliferation and differentiation. (**E**) The PI3K/AKT signaling pathway can be activated by MTA, Biodentine, iRoot BP Plus, Sr, and IL-6. Upon triggering, PI3K activates and generates PIP3, which, in turn, activates downstream AKT and mTOR signaling molecules. The PI3K/AKT pathway plays a role in anti-inflammatory processes by regulating apoptosis and proliferation and also promotes angiogenesis, cellular proliferation, and differentiation through interactions with other pathways.

## 6. Inflammatory Response Induced by BMs in VPT (Table 2)

### 6.1. MTA

#### 6.1.1. Effects of MTA on Inflammatory Factors

It is known that MTA contributes to the long-term reduction of pulpal inflammation and guides the restoration of pulpal tissue [79]. Santos et al. performed total pulpotomy using MTA and Biodentine on five beagles after one week of dentin exposure and took samples for observation after 14 weeks. The results demonstrated a substantial regenerative capacity of the pulp during the long-term restorative process, even in the presence of prior inflammatory conditions [80]. However, numerous studies have demonstrated that MTA’s effects on inflammation are not consistent throughout the inflammatory process, indicating that MTA does not exhibit anti-inflammatory activity at all stages of inflammation [14,81]. IL-6 and IL-8 were used to indicate the severity of inflammation [82]. Minsun Chung et al. [14] observed that after treating inflamed DPSCs with White MTA for 48 h, the expression levels of *IL-6* and *IL-8* significantly increased. However, another study reported that following 48 h of LPS treatment, the two markers decreased upon stimulation with Retro MTA [81]. Ciasca et al. observed that the treatment of inflamed *Human* osteoblast-like cells with ProRoot MTA resulted in the downregulation of *IL-1β* and *IL-6* within 48 h, and a gradual reduction of the inhibitory effect on *IL-6* was noted [83]. Nevertheless, it has been demonstrated that IL-6 secretion was not markedly enhanced or suppressed by MTA treatment of *Human* monocytes for 24 h, despite the notable downregulation of IL-1β [84]. These findings indicate that MTA induces varying inflammatory responses in different cell types. Even within the same cell type, different formulations of MTA elicit distinct inflammatory reactions. In addition, MTA may exert anti-inflammatory or pro-inflammatory effects that are subject to dynamic adjustment according to the time of action throughout the repair process.

**Table 2 biomolecules-15-00258-t002:** The impact of bioceramic materials on normal or inflamed cells of different types.

Materials	Target Cell	Cell State	References	Duration	Mechanism of Action	Quality
MTA	*Human* dental pulp stem cells (hDPSCs)	Normal	Minsun Chung et al. [81] (2019)	12~48 h	*IL-6*(+); *IL-8*(+);	High
			WY Lai et al. [85] (2013)		IL-1β(+), increasing with culturing time	High
			Min-Ching Wang et al. [14] (2023)	48~192 h	*IL-6*(+); *IL*-8(+), decreasing with culturing time	High
			Qiu et al. [86] (2017)	24 h	Activated Notch pathway via inhibiting autophagic	High
			J.M. Kim et al. [87] (2017)		Activated phospholipase C pathway	High
			Huang et al. [88] (2014)	1~5 d	Activate p38 pathway	High
			Kim et al. [89] (2020)	3~7 d	Initiated autophagy with AMPK activation	High
			Chen et al. [90] (2016)	1~7 d	Activated Wnt/β-catenin pathway	High
			Chen et al. [91] (2019)	1~14 d	CaSR activated phosphoinositide 3-kinase/Akt pathway	High
			J.-H. Kim et al. [92] (2019)	/	Combination of MTA and Propolis activated ERK pathway	High
			Woo et al. [93] (2015)	7 d	Combination of MTA and Platelet-Rich Fibrin activated BMP/Smad pathway	High
			Yun et al. [94] (2015)	7~14 d	Combination of MTA and Growth Hormone activated BMP and MAPK pathways	High
		Inflammatory	Minsun Chung et al. [81] (2019)	>48 h	*IL-6*(−); *IL-8*(−)	High
			Min-Ching Wang et al. [14] (2023)	48~192 h	*IL-6*(+) at 48 h and 96 h; *IL-8*(+)	High
			Wang et al. [95] (2021)	24~72 h	Activated AKT pathway	High
	*Human* neutrophils	Normal	Cavalcanti BN et al. [96] (2011)	48 h	IL-1β(+); IL-8(+)	Moderate
	*Human* monocytes	Normal	Alqassab et al. [84] (2023)	24 h	IL-1β(−); TNF-α(+)	Low
		Inflammatory			IL-1β(−); TNF-α(+)	
	*Human* osteoblast-like cells (MG-63)	Normal	Ciasca et al. [83] (2012)	24 h	*IL-1β*(±); *IL-6*(−); *IL-8*(+); *TNF-α*(−)	Moderate
				36 h	*IL-1β*(+); *IL-6*(±); *IL-8*(+); *TNF-α*(−)	
				48 h	*IL-1β*(+); *IL-6*(±); *IL-8*(+); *TNF-α*(±)	
		Inflammatory		24~48 h	*IL-1β*(−); *IL-6*(−); *TNF-α*(±); *IL-8*(−) at 24 h, *IL-8*(±) at 36~48 h	
	THP-1; *Human* microvascular endothelial cell line-1	Normal	Yeh et al. [97] (2018)	24~72 h	Induced macrophage polarization toward M2 phenotype, with upregulation of IL-10, TGF-β, and VEGF via Axl/Akt/NF-kB pathway	High
	Stem cells from apical papilla	Normal	Du et al. [98] (2020)	5 d	Activated p38 and ERK pathways	High
			Yan et al. [99] (2014)	3~7 d	Activated NF-κB pathway	High
	*Human* periodontal ligament stem cells	Normal	Wang et al. [100] (2017)	3~7 d	Activated NF-κB and MAPK pathways	Moderate
	RAW264.7 macrophage cells	Normal	X. Zhu et al. [101] (2017)	24 h	IL-1β(+); TNF-α(+); IL-10(+)	High
			Yuan et al. [102] (2018)		IL-1β(+); TNF-α(+); IL-6(+)	High
			M.-G. Tu et al. [103] (2019)		Induced autophagy	High
		Inflammatory	Kuramoto et al. [104] (2020)	1~5 h	*IL-10*(+); *Socs3*(+); Inhibited NF-κB activity and decreased *IL-1α* and *IL-6* via calcineurin/NFAT/Egr2 pathway	High
			Yuan et al. [102] (2018)	24 h	IL-1β(+); TNF-α(+); IL-6(+)	High
	L929 *Mouse* fibroblasts	Normal	Gomes-Filho et al. [105] (2009)	24 h	IL-1β(+); IL-8(±)	Low
	*Rat* bone marrow stromal cells	Normal	Y. Wang et al. [106] (2014)	3~7 d	Activated JNK and ERK MAPK pathways	Moderate
	*Rat* dental pulp cells	Inflammatory	Y Wang et al. [100] (2014)	3~14 d	Activated NF-κB pathway	High
Biodentine	hDPSCs	Normal	Jung et al. [107] (2014)	24~48 h	Activated MAPK pathway	High
			Min-Ching Wang et al. [14] (2023)	48~192 h	*IL-6*(−); ALP(−); *IL-8*(+)	High
			Luo et al. [108] (2014)	14 d	Activated MAPK and CaMKII pathways	High
		Inflammatory	Min-Ching Wang et al. [14] (2023)	48~192 h	*IL-6*(+) at 48 h, *IL-6*(−) at 192h; *IL-8*(+); ALP(−)	High
			Duaa Abuarqoub et al. [109] (2022)	24 h	Mediated polarization of M1 to M2 macrophages and enhanced anti-inflammatory cytokines	High
			Wang et al. [95] (2021)	24~72 h	Activated AKT pathway	High
	*Human* monocytes	Normal	Alqassab et al. [84] (2023)	24 h	IL-1β(+); TNF-α(+)	Moderate
		Inflammatory			IL-1β(−);TNF-α(+)	
	*Human* dental pulp fibroblasts	Inflammatory	Giraud et al. [110] (2017)	30 min	C5a secretion:Xeno III > TheraCal > Biodentine, Control; decrease THP-1 migration	Low
iRoot BP Plus	hDPSCs	Normal	Zhang et al. [111] (2015)	1 h	Activated ERK 1/2, JNK, and Akt pathways	High
		Inflammatory	Zeng et al. [112] (2023)	24 h	*TNF-α*(−); *IL-1β*(−); *IL-4*(+); *IL-6*(−); *IL-10*(+)	High
	Bone marrow mesenchymal stem cells	Normal	Lu et al. [113] (2019)	15~60 min	Activated MAPK pathway and autophagy	Moderate

Moreover, MTA materials themselves may trigger inflammation, and this pro-inflammatory effect becomes more pronounced over time. Some studies have investigated the effect of MTA on healthy DPSCs [14,81]. The results demonstrated that the pro-inflammatory marker *IL-1β* exhibited a significant increase within two days, while *IL-6* and *IL-8* demonstrated varying degrees of upregulation throughout the eight-day observation period. Additionally, the osteogenic marker ALP demonstrated a notable suppression. While a separate study demonstrated that MTA suppressed IL-1β expression in *Human* monocytes, this inhibitory effect was less pronounced than that observed in the inflammatory environment of a concurrent experiment [84]. These findings indicate that in the absence of an inflammation environment, MTA materials may induce inflammation and even inhibit mineralization. This propensity is likewise observed in *Human* neutrophils [96], *Human* fibroblasts [105], *Human* osteoblast-like cells [83], murine RAW264.7 macrophage cells [101], and L929 *Mouse* fibroblasts [105]. The fate determination of DPSCs plays a crucial role in their future development, which, in turn, influences the success of pulp-capping repair in clinical practice. The pro-inflammatory and mineralization-inhibitory effects of MTA on healthy DPSCs cannot fully explain its clinical application in indirect pulp capping, which aims to promote the formation of reparative dentin. The dentin mineralization-promoting effect of MTA in indirect pulp capping appears to be attributed to its influence on autophagy in hDPSCs.

#### 6.1.2. Effects of MTA on Cellular Autophagy

Two studies regarding the effects of MTA on autophagy in healthy *Human* hDPSCs indicate that MTA’s influence on cellular autophagy varies at different stages of action. Qiu et al. observed that MTA promoted cell proliferation and inhibited differentiation through the early inhibition of autophagy and activation of the Notch pathway within 24 h [86]. MTA may enhance the repair of damaged pulp by potentially accelerating the proliferation of hDPSCs and shortening the duration needed for these cells to progress into the odontoblastic differentiation phase in clinical practice. However, Kim et al. found that MTA promoted autophagy through the AMPK pathway and induced the differentiation and mineralization of adult dentin cells on days 3, 5, and 7 [89]. These results indicate a notable shift in the effect of MTA on autophagy, from initial inhibition to subsequent promotion by the second day. This change may be attributed to the activation of different signaling pathways in varying cellular microenvironments, leading to distinct effects on autophagy and potentially resembling a relay mechanism that promotes the proliferation, differentiation, and mineralization of DPSCs. Additionally, a study on MTA-induced murine healthy RAW264.7 macrophages demonstrated that cellular autophagy could be induced within 24 h, which is inconsistent with previous research [103]. This discrepancy may be attributed to differences in autophagy regulatory mechanisms across species. However, there is currently no relevant research on inflammatory DPSCs or other *Human* cells.

#### 6.1.3. Effects of MTA on Molecular Signaling Pathways

MTA has been observed to induce the activation of signaling pathways in DPSCs. In the absence of inflammation, the activity of the Akt [114], Phospholipase C [87], and Wnt [90] pathways can be observed following the treatment of cells with MTA for varying periods, from one day to two weeks. The regulation of pulpal inflammation and tooth repair by MTA is significantly influenced by the high involvement of the Ca sensing receptor (CaSR) and transient receptor potential ankyrin subfamily member 1 (TRPA1). Chen et al. demonstrated that CaSR is expressed in *Human* dental pulp. It was also shown that CaSR can negatively or positively regulate the MTA-induced mineralization of hDPSCs in a ligand-dependent manner via the phosphoinositide 3-kinase/Akt pathway [114]. J. M. Kim et al. conducted further studies on the relationship between the CaSR and MTA and found that MTA dually regulates extracellular Ca^2+^ and pH, activating the CaSR and subsequently activating multiple downstream pathways. Among these, Ca^2+^ mobilization from intracellular stores by the phospholipase C pathway plays an important role in the osteogenic differentiation of hDPSCs by regulating transcriptional activity [87]. CaSR mainly senses changes in Ca^2+^, while TRPA1 is the pathway by which odontogenic cells detect pH in the extracellular environment. The findings of Kimura et al. indicate that high pH stimulation results in the activation of intracellular Ca^2+^ mobilization via TRPA1 channel-mediated extracellular Ca^2+^ influx and intracellular Ca^2+^ release. Furthermore, under pathological conditions, TRPA1 channel activation directly promotes dentin formation [91]. In addition to the CaSR and TRAP1, Chen et al. also cultured hDPSCs using a range of concentrations of MTA extracts to examine their proliferation and odontogenic differentiation [90]. Their findings indicated that when hDPSCs were cultured in a wide range of concentrations of MTA extracts, genes, and proteins related to the Wnt/β-catenin signaling pathway were significantly elevated. This suggests that Wnt/β-catenin signaling is also involved in the odontogenic differentiation of hDPSCs.

Moreover, the MAPK pathway has been found to be the most frequently induced by MTA for pulpal osteogenic/odontogenic differentiation. J.-H. Kim et al. demonstrated that the treatment of hDPSCs with MTA and propolis, either alone or in combination, resulted in the phosphorylation of extracellular signal-regulated kinase (ERK) and the upregulated expression of dentin sialophosphoprotein (DSPP) and dental matrix protein 1 (DMP1) [92]. All three subfamily proteins of MAPK signaling (ERK, p38, and JNK) are targets of MTA for the promotion of dentin repair [88,98,106,107]. In addition, Du et al. [98] and Yan et al. [99] used MTA to co-culture with *Human* dental stem cells from apical papilla (hSCAPs) for periods ranging from three to seven days. The results demonstrated that different concentrations of MTA could promote the odontogenic/osteogenic differentiation potential of hSCAPs through the activation of the p38, ERK, or NF-κB signaling pathways. Furthermore, the NF-κB pathway was activated through the upregulation of inflammatory cytokines. Similarly, Wang et al. [100] observed the activation of the MAPK and NF-κB pathways in *Human* periodontal ligament stem cells.

Additionally, the combined use of MTA and platelet-rich fibrin (PRF) has been shown to synergistically promote the differentiation of hDPSCs into odontoblasts by regulating the bone morphogenetic protein (BMP)/Smad signaling pathway [93]. Yun et al. found that the co-administration of MTA and growth hormone could enhance the secretion of BMP2 and *p-Smad1/5/8* [94].

However, only a limited number of studies have investigated the activation of signaling pathways in inflammatory hDPSCs induced by MTA. Wang et al. [95] demonstrated that MTA enhanced the LPS-induced proliferation, adhesion, and differentiation of hDPSCs, with the proliferation and adhesion processes occurring via the AKT pathway. However, it is possible that the cell differentiation process may not utilize the same pathway. Previous studies have indicated that the differentiation process of inflammatory hDPSCs may be achieved through the activation of the NF-κB pathway. This is because MTA also has a certain pro-inflammatory tendency when it activates the pathway by acting on healthy pulp tissues. In a study conducted by Y. Wang et al. [100], *Rat* DPSCs were used to investigate the effects of MTA on tooth/osteogenic capacity. The findings indicated that MTA enhanced this capacity at the inflammatory site by activating the NF-κB pathway, which indirectly confirmed the hypothesis.

It is noteworthy that Kuramoto et al. [104] discovered that MTA inhibited NF-κB activity and decreased *IL-1α* and *IL-6* via the calcineurin/NFAT/Egr2 pathway when inflammatory RAW264.7 macrophage cells were incubated with MTA for a period of 5 h. This suggests that the modulatory effects of MTA on certain signaling pathways, such as the NF-κB pathway, may be dynamic in the context of an inflammatory environment. This is in contrast to the findings of most studies, which report a single activating or inhibitory effect. Rather, the effects of MTA on signaling pathways may be context-dependent and vary with the development of inflammation and alterations in the microenvironment. Consistent with previous studies, Y. Wang et al. [108] also discovered that MTA could enhance odontogenic and osteogenic capacity through the activation of the JNK and ERK pathways following the treatment of healthy *Rat* bone marrow stromal cells with MTA for one week.

#### 6.1.4. Effects of MTA on Macrophages

Furthermore, studies have demonstrated that MTA induced macrophage polarization towards the M2 phenotype, increasing the secretion of IL-10, TGF-β, and VEGF through the Axl/Akt/NF-kB pathway, which, in turn, exerts significant anti-inflammatory effects [97]. This process is associated with a microenvironment of high pH and the gradual release of calcium ions from MTA.

### 6.2. Biodentine

Similar to MTA, Biodentine can promote the odontogenic/osteogenic differentiation of dental pulp through the MAPK [107] and AKT [95] pathways. Additionally, Luo et al. [108] discovered that Biodentine also plays a role in inducing odontogenic/osteogenic differentiation through the calcium-/calmodulin-dependent protein kinase II (CaMKII) signaling pathway, where CaMKII facilitates its induction by promoting the phosphorylation of *Smad1* [115].

Currently, numerous studies have investigated the effects of Biodentine on the pulp inflammation response. In healthy hDPSCs, Biodentine inhibited IL-6 secretion for up to 192 h, with a progressive increase in inhibition over time. However, in the context of an inflammatory state, Biodentine unexpectedly promoted *IL-6* secretion during the initial 48 h. Nevertheless, the inhibitory effect was observed to resume from the 96 h mark onwards [14]. This pattern is notably distinct from the observed tendency of MTA to moderately promote inflammatory responses in healthy cells at the early stages and to suppress these responses in inflammatory cells thereafter. Furthermore, two additional studies demonstrated that Biodentine consistently promoted *IL-8* secretion in both inflammatory and non-inflammatory states throughout the entire eight-day observation period [14,81]. Previously, it was assumed that both IL-6 and IL-8 were regarded as inflammatory markers. However, the results of these studies indicate that both cytokines were not simultaneously up- or downregulated. The variations in the induction of IL-6 and IL-8 may suggest that cellular inflammation during different phases is regulated by distinct immune cell populations. Additionally, in the case of IL-6, Biodentine did not simply promote or inhibit its secretion. This suggests that the multifunctionality of the inflammatory cytokines in pulpal inflammation and the effect of Biodentine on them are also dynamically adjusted, similar to the effects observed with MTA.

Although there is a substantial body of literature indicating that complement, particularly C3a fragments and C5a fragments, plays a significant role in the initiation of pulpal inflammation and the subsequent reparative regeneration of damaged pulp, few studies have examined the impact of bioceramic materials on complement secretion. A study utilizing Biodentine, TheraCal, and Xeno Ⅲ to incubate injured pulp fibroblasts for 30 min demonstrated that Biodentine had no significant effect on C5a secretion, whereas TheraCal and Xeno Ⅲ, which contain resin components, significantly promoted C5a secretion, with the latter exhibiting a more pronounced effect [110]. Notably, C5a secretion was positively correlated with resin content. This phenomenon can be attributed to the more severe inflammatory response caused by the lower biocompatibility of the resin. In contrast, Biodentine demonstrated no significant promotion or inhibition of C5a secretion in inflammatory conditions, suggesting that the active ingredient in the calcium silicate material may not affect pulpal inflammation through its filling properties.

A study investigating Biodentine in the treatment of LPS-stimulated *Human* macrophages for a period of 24 h observed a reduction in the secretion of pro-inflammatory cytokines IL-1β, IL-6, and IL-8, accompanied by an increase in the secretion of anti-inflammatory cytokines IL-10 and TGF-β [109]. This finding suggests that Biodentine may contribute to the polarization of macrophages from the M1 phenotype to the M2 phenotype.

### 6.3. iRoot BP Plus

Studies indicate that iRoot BP Plus facilitates the pulp–dentin complex repair pathway, which is comparable to MTA. Zhang et al. [111] demonstrated that iRoot BP Plus facilitates hDPSCs in their migration and pulp repair through the FGFR-mediated ERK 1/2, JNK, and Akt pathways. Lu et al. [113] found iRoot BP Plus enhanced the bone-derived/odontogenic differentiation potential of BMSCs via the MAPK pathway.

A study utilizing iRoot BP Plus to treat hDPSCs in an inflammatory state for 24 h revealed a reduction in the secretion of pro-inflammatory factors *IL-1β* and *IL-6*, accompanied by an increase in the secretion of anti-inflammatory factors *IL-4* and *IL-10*. This finding suggests that iRoot BP Plus is capable of inhibiting inflammation in a relatively short period of time [112].

In addition to the aforementioned studies on MTA, there are also investigations using iRoot BP Plus on bone marrow mesenchymal stem cells (BMSCs) [113]. The results demonstrated that autophagy markers were progressively upregulated at 15, 30, and 60 min, indicating that iRoot BP Plus is capable of promoting the bone/odontogenic differentiation of BMSCs through autophagy and that it induces cellular autophagy in a relatively short period of time.

### 6.4. Biologically Active Ions (Table 3)

Biologically active ions are defined as those ions that can interact with biological systems and have an effect on biological processes. Among the most studied ions are calcium, iron, silicon, zinc, magnesium, lithium, silver, phosphorus, and strontium. These ions have been shown to promote bone regeneration and tissue repair. Besides the main active components of calcium silicate materials—calcium, silicon, iron, and phosphorus—which have been introduced in the earlier sections, this part will focus on the effects of other active ions on the inflammatory response.

**Table 3 biomolecules-15-00258-t003:** The impact of bioactive ions on normal or inflamed cells of different types.

Materials	Material Forms	Target Cell	Cell State	References	Duration	Mechanism of Action	Quality
Lithium	Lithium-doped mesoporous nanoparticles	hDPSCs	Normal	Liang et al. [116] (2023)	1~7 d	Activated Wnt/β-catenin pathway	High
	Phosphate- and borate-based bioactive glasses that contained lithium	*Mouse* dental pulp cell		Zhang et al. [117] (2019)	24 h		High
	Lithium-containing bioactive glass	CD1 wild-type *Mice*	Exposed pulp	Alaohali et al. [118] (2021)	1 d		Low
Zinc	Zinc ions; zinc-containing bioactive glasses (ZnBGs)	hDPSCs	Normal	Huang et al. [119] (2017)	1~10 d	Zn ions enhanced proliferation; ZnBGs increased DSPP and DMP-1	High
	Calcium phosphate cements (CPCs) incorporating ZnBG			Zhang et al. [120] (2014)	7~14 d	Activated integrin, Wnt, MAPK, and NF-kB pathways	High
Strontium	SrCl2.6H2O	hDPSCs	Normal	M. Huang et al. [121] (2016)	7~14 d	Increased DSPP and DMP-1 via CaSR pathway	High
	Strontium (Sr^2+^); Borate (BO_3_^3−^); Silicate (SiO_3_^2−^)			Y. Miyano et al. [122] (2022)	14~28 d	Induced differentiation into odontoblast-like cells	High
	Strontium ranelate (SrRn)	*Mouse* dental papillae cells		Bakhit et al. [123] (2018)	1~7 d	Activated PI3K/Akt pathway via CaSR	High
Magnesium	MgCl_2_	hDPSCs	Normal	Kong et al. [124] (2019)	1~3 d	Activated ERK/BMP2/Smads pathway	High
	Magnesium-doped bioactive glass			Y Zhong et al. [125] (2024)	7~14 d	Upregulated DSSP, DMP-1, ALP, OCN, and RUNX2; suppressed *IL-4*, *IL-6*, *IL-8*, and *TNF-α*	High
Silver	Silver-doped bioactive glass/chitosan (Ag-BG/CS)	hDPSCs	Inflammatory	N Zhu et al. [126] (2019)	24 h	Activated MAPK pathway	High
				N. Zhu, et al. [127] (2019)	7 d	Downregulated *IL-1β*, *IL-6*, *IL-8*, *TNF-α* via inhibiting NF-κB pathway	High

Currently, the literature on the use of bioactive ions for targeted studies on pulpal inflammation remains limited. Nonetheless, we have identified numerous studies that elucidate the relationship between bioactive ions and odontogenic repair in pulpal treatment. Therefore, we will provide a detailed summary and discussion of these findings.

#### 6.4.1. Lithium Ions

Lithium ions (Li^+^), as antagonists of glycogen synthase kinase 3 (GSK3), can mitigate the inhibitory effects of GSK3 on the Wnt signaling pathway, thereby indirectly activating the Wnt pathway. Through this mechanism, lithium ions modulate pulpal inflammation and promote pulp repair and healing [117]. Liang et al. synthesized lithium-doped mesoporous nanoparticles (Li-MNPs) to treat hDPSCs. The results demonstrated that Li-MNPs significantly enhanced mineralization and odontogenic differentiation, thereby promoting dentin regeneration both in situ and in vivo [116]. Furthermore, Alaohali et al. replaced sodium ions with lithium in BGs and observed tertiary dentin formation in pulp-capping experiments [118]. Ishimoto et al. employed LiCl for pulp capping and observed the formation of tubular dentin [128].

#### 6.4.2. Zinc Ions

Zinc ion (Zn^2+^) compounds like zinc oxide and clove oil have long been used in dental treatments for endodontic diseases. Huang et al. prepared zinc and zinc-containing bioactive glasses (ZnBGs) to treat hDPSCs. The results demonstrated that ZnBG increased the secretion of DSPP and DMP-1, as well as upregulated the mRNA of osteogenic markers and the expression of vascular endothelial growth factor (VEGF) [119]. Zhang et al. prepared a bioactive calcium phosphate cement (CPC) containing ZnBG by a sol-gel process and investigated its effects on hDPSCs, demonstrating the activation of odontogenic differentiation and the promotion of angiogenesis via the integrin, Wnt, MAPK, and NF-κB pathways [120]. Among them, integrins, especially integrin α5 and α6, play a pivotal role in the proliferation, migration, and osteogenic/dentinogenic differentiation of hDPSCs [129,130].

#### 6.4.3. Strontium Ions

Strontium ions (Sr^2+^) can replace calcium ions in enamel, enhancing the hardness of enamel, improving the structure and function of dentin, and improving the blood circulation of pulpal tissues. Bakhit et al. employed strontium ranelate (SrRn) to *Mouse* dental pulp cells (MDPs). Their findings demonstrated that SrRn stimulates the proliferation of MDPs and tooth formation via CaSR-activated PI3K/Akt signaling in vitro and induces osteogenic differentiation and mineralization [123]. The results of another experiment demonstrated that Sr^2+^ may induce hDPSCs to differentiate into dentinogenic cell-like cells [122]. Additionally, Huang et al. demonstrated that a specific dose of Sr could promote the proliferation, odontogenic differentiation, and mineralization of hDPSCs in vitro through the CaSR activation of the downstream MAPK/ERK pathway [121].

#### 6.4.4. Magnesium Ions

Magnesium ions (Mg^2+^) primarily function in dentin and cementum. However, recent studies have demonstrated that Mg^2+^ also functions in pulp. Kong et al. found that a Mg^2+^-enriched microenvironment activated the ERK/BMP2/Smads signaling pathway, promoting the odontogenic differentiation of DPSCs [124]. Zhong et al. synthesized Mg-BG to investigate its effects on mineralization, tooth formation, and the anti-inflammatory capacity of hDPSCs. The results revealed an increase in the expression of odontogenic genes, accompanied by a downregulation of inflammatory markers, including *IL-4*, *IL-6*, *IL-8*, and *TNF-α* [125].

#### 6.4.5. Silver Ions

In addition to the antimicrobial and sealing of dentin tubules functions, silver ions (Ag^+^) have been shown to have a positive impact on modulating pulpal inflammation and promoting pulpal repair and healing. Zhu et al. introduced silver-doped BG to chitosan hydrogel (Ag-BG/CS) and applied it to inflamed DPSCs and *Rat* inflamed dental pulp models [126,127]. The results of cellular experiments demonstrated that Ag-BG/CS downregulated the expression of *IL-1β*, *IL-6*, *IL-8*, and *TNF-α* by inhibiting the NF-κB pathway and enhanced the in vitro odontogenic differentiation potential of DPSCs. In vivo experiments further indicated that Ag-BG/CS enhanced the preservation of vital pulp tissue and induced stronger restorative dentin formation compared with MTA. Additionally, the significantly increasing phosphorylation levels of p38 and ERK1/2 suggested that Ag-BG/CS enhances pulpal restoration through the MAPK signaling pathway.

### 6.5. Bioactive Proteins (Table 4)

It is well established that the signaling pathways involved in the repair and healing of the pulp–dentin complex require the transmission or reception of information through a diverse array of proteins. This has inspired researchers to apply these proteins directly for the regulation of this process. In addition, there are proteins such as VEGF that have a direct contribution to pulp repair and healing. In this paper, we collectively refer to them as bioactive proteins.

**Table 4 biomolecules-15-00258-t004:** The impact of bioactive proteins on normal or inflamed cells of different types.

Materials	Target Cell	Target State	References	Duration	Mechanism of Action	Quality
Chitosan biguanidine loaded with VEGF and BMP-2	hDPSCs	Normal	B. Divband et al. [131] (2021)	7 d	ALP(+); Collagen-1(+); OCN(+)	High
BMP-7			S.K. Eren et al. [132] (2022)	21 d	Combination of BMP-7 and MTA increased odontogenic/osteogenic differentiation	High
			Liang et al. [133] (2022)		Induced migration and angiogenesis	High
BMP-9		Inflammatory	Song et al. [134] (2022)	/	IL-6(−); IL-8(−); CCL2(−); MMP2(−); p-Smad1/5(+); p-ERK(−); p-JNK(−)	High
Heparin		Normal	Rodrigues et al. [135] (2019)	1~3 d	*BMP-2*(+); *OCN*(+); induced osteogenic bioactivity	High
Treated dentin matrix (TDM); small extracellular vesicle (sEV)		Normal	Wen et al. [136] (2020)	7 d	The sEV promoted proliferation and migration. Combination promoted migration, suppressed proliferation, enhanced odontoblast-related protein expression	High
Poly Aspartic Acid	Wistar *Rats* (250–300 g)	Exposed pulp	dos Santos et al. [137] (2023)	7~21 d	DMP-1(+)	Low
Resolvin E1	SD *Rats* (8-week-old)	Exposed pulp	Chen et al. [138] (2021)	1~28 d	DMP-1(+); DSPP(+)	Low

The most extensively studied bioactive proteins are the BMP family. A study revealed a live pulpotomy in experimental dogs applying BMP-2 and BMP-4 in combination with dentin matrix by Nakashima, indicating significant promotion in dentin formation and suggesting the potential to induce differentiation [139]. Another study found that combining BMP-2 with VEGF significantly enhances the proliferation of hDPSCs [131]. These findings indicate that BMP-2 may enhance the proliferation and differentiation abilities of DPSCs. A study found that BMP-7, in combination with MTA, had no significant effect on cell proliferation compared with MTA alone when treating DPSCs. However, an increase in mineralized nodules and a high expression of *DMP-1* and *DSPP* were observed [132]. Moreover, Liang et al. demonstrated that BMP-7 promoted the migration and odontogenic differentiation of hDPSCs [133]. In conclusion, BMP-7 appears to promote the migration of DPSCs rather than proliferation. BMP-9 is frequently associated with inflammatory responses, yet its relationship to endodontitis remains unclear. Song et al. studied BMP-9 expression in *Rats* with endodontic inflammation and in immortalized hDPSCs, using THP-1 to assess BMP-9’s role. They found that BMP-9 overexpression decreased IL-6 and matrix metalloproteinase 2 (MMP-2) secretion, increased phosphorylated Smad1/5, and reduced phosphorylated ERK and JNK levels. BMP-9 also reduced THP-1 cell migration [134]. These findings indicate that BMP-9 may play a role in the early stages of inflammation, exerting a partial inhibitory effect on its severity.

Resolvin E1 is synthesized during the spontaneous regression phase of acute inflammation. Chen et al. utilized 8-week-old SD *Rats* to model pulp injury and sealed the pulp with collagen sponges impregnated with Resolvin E1 for a period of 4 weeks [138]. The results demonstrated enhanced DMP-1 and DSPP secretion, as well as restorative dentin formation.

Furthermore, the effect of heparin on hDPSCs has been investigated, and it has been found that heparin induces osteogenic bioactivity and increases *BMP-2* and *osteocalcin (OCN)* [135]. The impact of a combination of dentin matrix proteins (TDM) and DPSC-derived small extracellular vesicles (sEVs) on the repair of pulp–dentin complexes has also been investigated [136]. The results demonstrated that sEVs enhanced the proliferation and migration of DPSCs. The combination of TDM and sEVs exhibited a synergistic effect on DPSC migration while simultaneously inhibiting their proliferation. In vivo TDM and sEV-TDM were observed to promote the formation of dentin, and odontoblast-like cells were observed. Furthermore, studies have been conducted showing that pAsp promotes the secretion of osteopontin (OPN) and DMP-1 and facilitates dentin regeneration in the absence of additional calcium sources in dentin regeneration [137].

## 7. Conclusions

A notable heterogeneity characterized the design of the studies incorporated within the scope of this review, encompassing in vitro cellular experiments on diverse cell types of *Human* and animal origin, along with animal model studies employing *Rats* or *Mice*. These variations have the potential to influence the comparison and integration of study results. Additionally, the experimental conditions (e.g., the employed inflammation model, the concentration of materials utilized) and outcome metrics (e.g., expression levels of inflammatory factors, biocompatibility assessment) exhibited significant variation across studies. Consequently, although the present analysis offers a comprehensive understanding of the role of bioactive materials in pulpal inflammation modulation and regeneration, readers should exercise caution when directly comparing different studies.

In conclusion, with the increasing use of living pulp preservation in clinical practice, it becomes particularly important to investigate the mechanisms that promote the repair and healing of pulp. Thus, the question of how to regulate non-long-term mild or moderate inflammation as a prerequisite for inducing pulp repair and healing has emerged as a significant area of investigation. In vivo inflammation is primarily expressed by pro- and anti-inflammatory cytokines, which are regulated by macrophages, complement, autophagy, and other factors. These cells or proteins regulate the relevant cytokines through signaling pathways, either directly or indirectly, thus altering the inflammatory environment of damaged pulp. This, in turn, initiates pulpal restoration, ultimately leading to pulp healing and dentin formation. To facilitate the regulation of this complex process, BMs that interact with biological systems and produce specific biological effects have emerged. This review initially dedicates a section to elucidate the effects of various inflammatory cytokines, signaling pathways, complement, autophagy, and macrophages on inflammation. Subsequently, the BMs are divided into calcium silicate materials, bioactive ions, and bioactive proteins, and their effects on the pulp–dentin complex are discussed individually.

The calcium silicate materials, such as MTA, Biodentine, and iRoot BP Plus, demonstrated comparable yet not identical effects on various inflammatory cytokines, including IL-1β, IL-6, TNF-α, and IL-8. Furthermore, the observed effects were not exclusively promotional or inhibitory. The complement C3a and C5a, which play a pivotal role in the initiation of inflammation, did not establish a necessary relationship with the active components of calcium silicate materials in this study. Instead, their expression was found to be positively correlated with the resin content of the material. In studies of autophagy, MTA’s influence on cellular autophagy varies at different stages of action, and it is clear that iRoot BP Plus is capable of activating autophagy. Regarding the effect of MTA and iRoot BP Plus on macrophages, it can be concluded that these materials promote M1/M2 phenotype polarization. Biodentine also appears to facilitate macrophage polarization from the M1 to the M2 phenotype. In terms of signaling pathways, in addition to the MAPK and AKT pathways, Biodentine also activates the CaMKII pathway. MTA is also linked to the NF-κB, BMP/Smad, Wnt-catenin, and CaSR pathways. All of the aforementioned signaling pathways have been shown to have a positive effect on the osteogenic/odontogenic differentiation of hDPSCs.

In addition to the calcium, silicon, and iron elements present in the active components of calcium silicate materials, bioactive ions—including lithium, zinc, strontium, magnesium, and silver—facilitate the reparative regeneration of the pulp–dentin complex. Among these, Li promotes tooth regeneration by activating the integrin, Wnt, MAPK, and NF-κB pathways, while Sr correlates with the PI3K/Akt and MAPK/ERK pathways. In Mg-rich microenvironments, ERK/BMP2/Smads signaling is activated by an increase in intracellular Mg^2+^. Ag activates ERK/BMP2/Smads signaling by inhibiting the NF-κB pathway and activating the MAPK pathway, which, in turn, promotes pulp repair. The most extensively studied bioactive proteins include BMP-2, BMP-7, and BMP-9 of the BMP protein family, in addition to heparin, TDM, and pAsp, all of which have been shown to promote tooth regeneration. However, their antimicrobial properties are slightly weaker than those of the other two BMs.

## 8. Perspective

A considerable number of BMs have been investigated with the objective of modulating pulpal inflammation and promoting the restorative healing of damaged pulp. However, despite this research, BMs remain the only materials used in clinical practice. Although they do achieve satisfactory results, they are still quite far from perfectly realizing VPT, namely the functional healing of the pulp and dentin. The principal active components of calcium silicate materials remain a limited number of elements, although an increasing number of modified BMs are being synthesized and studied. Different BMs elicit varying inflammatory responses in dental pulp due to their distinct compositions. Understanding their modes of action will contribute to a broader understanding of the mechanisms involved in induced dentinogenesis. This feature provides clinicians with greater options to select the most appropriate BMs for precision therapy based on the patient’s specific case status. Furthermore, the currently widely used clinical BMs, including MTA, Biodentine, and iRoot BP Plus, have demonstrated efficacy in endodontic restorative healing and hard tissue generation. It is anticipated that the therapeutic effect of VPT will be further enhanced in the future through the incorporation of novel biologically active ions, such as Li and Ag, along with bioactive proteins, including the BMP family. Furthermore, we aspire to promote the advancement of the integration of materials science with immunology, molecular biology, and other fields by discussing the dynamic inflammatory regulation of BMs in VPT. This integration is expected to result in the development of smarter, novel materials that can sense and respond to changes in the inflammatory microenvironment in real time, thereby enhancing their role in therapy.

These insights will help in the development of new materials with specific components, aiding in exploring how newly developed pulp-capping materials shift the balance from inflammation toward repair and healing. This understanding will also lay the foundation for designing future capping materials aimed at influencing these processes. In the future, it may be possible to gradually approach the goal of perfectly realizing VPT through the organic combination of a more diverse range of bioactive ions and bioactive proteins.

## Figures and Tables

**Figure 1 biomolecules-15-00258-f001:**
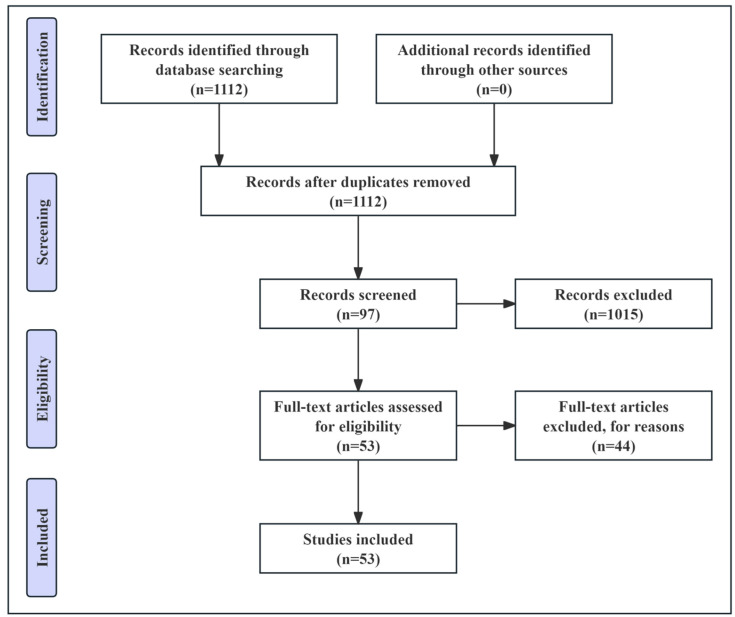
The initial search results.

**Table 1 biomolecules-15-00258-t001:** The initial search results.

Search Terms	Search Results
Protein + dental pulp + inflammation	561
Lithium/Zinc/Strontium/Magnesium/Silver + dental pulp	61
Mineral trioxide aggregate/Biodentine/iRoot BP Plus + inflammation/signaling pathway/molecular mechanism/vital pulp therapy	490
Total	1112

## Data Availability

Not applicable.

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
