# Peer review of "Bioactive Materials in Vital Pulp Therapy: Promoting Dental Pulp Repair Through Inflammation Modulation"

_biomolecules, 2025, doi:10.3390/biom15020258_

Round 1
Reviewer 1 Report
Comments and Suggestions for Authors
The goal of this comprehensive literature review is to explore the mechanisms through which biomaterials regulate the balance between tissue inflammation and regeneration. The manuscript is well-structured, with a title that accurately reflects the content. The information presented is relevant to the field.
To enhance the rigor and scientific value of the manuscript, some areas require attention. First, the legend accompanying Figure 1 should be supported by adequate references to previous studies that validate the statements presented. This will strengthen the scientific foundation of the figure and ensure it aligns with the existing body of literature. Including these references will enhance the credibility and robustness of the manuscript.
Additionally, the introduction would benefit from addressing recent studies that explore the impact of pre-operative pulp inflammation on pulpotomy outcomes. This discussion would reinforce the idea that pulp has a significant regenerative potential, even in the presence of prior inflammatory conditions. Including insights from studies such as “Influence of preoperative pulp inflammation in the outcome of full pulpotomy using a dog model” (https://doi.org/10.1016/j.joen.2021.06.018) would provide valuable context and strengthen the manuscript’s argument. This addition would also highlight the evolving understanding in the field and align the manuscript with contemporary scientific advancements.
While the presentation of the results is clear, a more thorough discussion of their clinical implications could further enhance the manuscript.
Comments on the Quality of English Language
Sentences structure can be improved.
Author Response
Reviewer #1: The goal of this comprehensive literature review is to explore the mechanisms through which biomaterials regulate the balance between tissue inflammation and regeneration. The manuscript is well-structured, with a title that accurately reflects the content. The information presented is relevant to the field.
Response: We sincerely thank you for the time and energy you expended on reviewing our paper. Your comments provide valuable insights to refine its contents and analysis. We try to address the issues raised as best as possible. In the manuscript, the revised text according to the reviewer’s comments is highlighted in yellow. In the following sections, you will find our responses to your points and suggestions.
Comments 1:To enhance the rigor and scientific value of the manuscript, some areas require First, the legend accompanying Figure 1 should be supported by adequate references to previous studies that validate the statements presented. This will strengthen the scientific foundation of the figure and ensure it aligns with the existing body of literature. Including these references will enhance the credibility and robustness of the manuscript.
Response 1: Thank you for the careful reviewing. In order to enhance the manuscript's rigor and scientific value of the manuscript, nine additional references have been incorporated. These references substantiate the assertions regarding the molecular mechanisms illustrated in Figure 1 (revised version: Figure 2) on page 7, as shown below:
“72. Artavanis-Tsakonas, S.; Matsuno, K.; Fortini, M.E. Notch signaling. Science 1995, 268, 225-232, doi:10.1126/science.7716513.
73. Cantley, L.C. The phosphoinositide 3-kinase pathway. Science 2002, 296, 1655-1657, doi:10.1126/science.296.5573.1655.
74. Shindo, K.; Kawashima, N.; Sakamoto, K.; Yamaguchi, A.; Umezawa, A.; Takagi, M.; Katsube, K.; Suda, H. Osteogenic differentiation of the mesenchymal progenitor cells, Kusa is suppressed by Notch signaling. Exp Cell Res 2003, 290, 370-380, doi:10.1016/s0014-4827(03)00349-5.
75. Thomas, G.M.; Huganir, R.L. MAPK cascade signalling and synaptic plasticity. Nat Rev Neurosci 2004, 5, 173-183, doi:10.1038/nrn1346.
76. Gaestel, M. MAPKAP kinases - MKs - two's company, three's a crowd. Nat Rev Mol Cell Biol 2006, 7, 120-130, doi:10.1038/nrm1834.
77. Anjum, R.; Blenis, J. The RSK family of kinases: emerging roles in cellular signalling. Nat Rev Mol Cell Biol 2008, 9, 747-758, doi:10.1038/nrm2509.
78. Bray, S.J. Notch signalling in context. Nat Rev Mol Cell Biol 2016, 17, 722-735, doi:10.1038/nrm.2016.94.
79. Yu, H.; Lin, L.; Zhang, Z.; Zhang, H.; Hu, H. Targeting NF-kappaB pathway for the therapy of diseases: mechanism and clinical study. Signal Transduct Target Ther 2020, 5, 209, doi:10.1038/s41392-020-00312-6.
80. Liu, J.; Xiao, Q.; Xiao, J.; Niu, C.; Li, Y.; Zhang, X.; Zhou, Z.; Shu, G.; Yin, G. Wnt/beta-catenin signalling: function, biological mechanisms, and therapeutic opportunities. Signal Transduct Target Ther 2022, 7, 3, doi:10.1038/s41392-021-00762-6.”
Comments 2:Additionally, the introduction would benefit from addressing recent studies that explore the impact of pre-operative pulp inflammation on pulpotomy outcomes. This discussion would reinforce the idea that pulp has a significant regenerative potential, even in the presence of prior inflammatory conditions. Including insights from studies such as “Influence of preoperative pulp inflammation in the outcome of full pulpotomy using a dog model” (https://doi.org/10.1016/j.joen.2021.06.018) would provide valuable context and strengthen the manuscript’s argument. This addition would also highlight the evolving understanding in the field and align the manuscript with contemporary scientific advancements.
Response 2: Thanks a lot for the careful reviewing. Your suggestions have been greatly helpful in improving the rigor of our manuscript. The study you provided was instrumental in conducting this review. We have included it in section VI: Inflammatory response induced by BMs in VPT, page 9, lines 299-303, as shown below:
“Santos et al. performed total pulpotomy using MTA and Biodentine on five Beagles after one week of dentin exposure and took samples for observation after 14 weeks. The results demonstrated a substantial regenerative capacity of the pulp during the long-term restorative process, even in the presence of prior inflammatory conditions [82].”
Comments 3:While the presentation of the results is clear, a more thorough discussion of their clinical implications could further enhance the manuscript.
Response 3: Thank you very much for your insightful suggestion. The guidance you have provided is quite pertinent. A discussion of the clinical significance and future applications of BMs has been included in section VIII: Perspective, page 18, lines 683-695, as shown below:
“This feature provides clinicians with greater options to select the most appropriate BMs for precision therapy based on the patient's specific case status. Furthermore, currently widely used clinical BMs, including MTA, Biodentine, and iRoot BP Plus, have demonstrated efficacy in endodontic restorative healing and hard tissue generation. It is anticipated that the therapeutic effect of VPT will be further enhanced in the future through the incorporation of novel biologically active ions, such as Li and Ag, along with bioactive proteins, including the BMP family. Furthermore, we aspire to promote the advancement of the integration of materials science with immunology, molecular biology, and other fields by discussing the dynamic inflammatory regulation of BMs in VPT. This integration is expected to result in the development of smarter, novel materials that can sense and respond to changes in the inflammatory microenvironment in real time, thereby enhancing their role in therapy.”

Reviewer 2 Report
Comments and Suggestions for Authors
This review article provides a comprehensive summary of the molecular mechanisms underlying the actions of currently available dental bioactive materials used in vital pulp therapy, including MTA and other calcium silicate-based materials. It highlights the properties of these materials, emphasizing their roles in promoting inflammation, as well as reducing inflammation and supporting tissue repair/healing. This article falls within the scope of the journal Biomolecules and may contribute to increase our understanding of the mechanisms by which dental bioactive materials promote repair/healing of the diseased/exposed dental pulp tissue.
Comments
1. Please provide additional details on how the literature search was performed. The abstract mentions that the search was conducted through the PubMed and MEDLINE databases, but this is not addressed in the main text. Please clarify the search terms used and the inclusion/exclusion criteria applied during the search.
2. "Healing" and "regeneration" are distinct concepts, and it is desirable to differentiate between them. Many of the studies extracted in this research demonstrated increased production of anti-inflammatory cytokines and enhanced expression of mineralization-related factors, which are mechanisms more closely associated with repair/healing rather than regeneration.
3. In Introduction (page 2), please include appropriate reference(s) after the following sentences:
- lines 55-57
- lines 58-60
- lines 71-74
4. Please spell out abbreviations where they first appear.
Author Response
To Reviewer #2: This review article provides a comprehensive summary of the molecular mechanisms underlying the actions of currently available dental bioactive materials used in vital pulp therapy, including MTA and other calcium silicate-based materials. It highlights the properties of these materials, emphasizing their roles in promoting inflammation, as well as reducing inflammation and supporting tissue repair/healing. This article falls within the scope of the journal Biomolecules and may contribute to increase our understanding of the mechanisms by which dental bioactive materials promote repair/healing of the diseased/exposed dental pulp tissue.
Response: We sincerely thank you for the time and energy you expended on reviewing our paper. Your comments provide valuable insights to refine its contents and analysis. We try to address the issues raised as best as possible. In the manuscript, the revised text according to the reviewer’s comments is highlighted in yellow. In the following sections, you will find our responses to your points and suggestions.
Comments 1:
Please provide additional details on how the literature search was performed. The abstract mentions that the search was conducted through the PubMed and MEDLINE databases, but this is not addressed in the main text. Please clarify the search terms used and the inclusion/exclusion criteria applied during the search.
Response 1: Thank you for the careful reviewing. The inclusion/exclusion criteria and search terms have been incorporated in Section II, page 2, as shown below:
“2.Inclusion and Exclusion Criteria
The inclusion criteria were as follows:
- Original articles
- Human or animal cell culture studies, or animal studies
The exclusion criteria were:
- Case study reports
- Review or systematic review
- Commentaries/letters to the editor/expert opinion
- Non–English language articles
Search Methodology
The MEDLINE/PubMed library databases were queried for relevant articles to the topic of application of bioactive materials in VPT published up to December 2023 (last accessed December 31, 2023). The search terms were the following key words used in various combinations: “Mineral trioxide aggregates”, “Biodentine”, “iRoot BP Plus”, “pulp capping”,“lithium”, “zinc”, “Strontium”, “Magnesium”, “Silver” “inflammation”, “molecular mechanism”, “signaling pathway”, “dental stem cell”, “apical papilla stem cell”, “dental pulp”.
An initial literature search using different combinations of the search terms yielded 1112 articles (Table 1). Figure 1 presents a flowchart of the review process. Titles and abstracts of these articles were reviewed by 2 independent examiners who excluded nonqualifying publications.”
Table 1. The initial search results.
|
Search Terms |
Search Results |
|
Protein + dental pulp + inflammation |
561 |
|
Lithium/Zinc/Strontium/Magnesium /Silver + dental pulp |
61 |
|
Mineral trioxide aggregate/Biodentine/iRoot BP Plus + inflammation/ signaling pathway/molecular mechanism/vital pulp therapy |
490 |
|
Total |
1112 |
Figure 1. The initial search results.
Comments 2: "Healing" and "regeneration" are distinct concepts, and it is desirable to differentiate between them. Many of the studies extracted in this research demonstrated increased production of anti-inflammatory cytokines and enhanced expression of mineralization-related factors, which are mechanisms more closely associated with repair/healing rather than regeneration.
Response 2: Thank you a lot for your suggestion. We have given your suggestion careful consideration, and we believe it to be accurate. In the section of the study discussing anti-inflammatory and mineralization processes, the term "regeneration" has been substituted with "healing."
Comments 3: In Introduction (page 2), please include appropriate reference(s) after the following sentences:
- lines 55-57
- lines 58-60
- lines 71-74
Response 3: Thanks for the careful reviewing. We have added the corresponding references following the three sentences, as shown below:
- lines 55-57: When inflammation is reduced to a low level, the tissue microenvironment changes, and the balance shifts towards restorative repair [10].
“10. Scheller, J.; Chalaris, A.; Schmidt-Arras, D.; Rose-John, S. The pro- and anti-inflammatory properties of the cytokine interleukin-6. Biochimica et Biophysica Acta (BBA) - Molecular Cell Research 2011, 1813, 878-888, doi:10.1016/j.bbamcr.2011.01.034.”
- lines 58-60: Dental pulp stem cells (DPSCs) can differentiate in multiple directions, facilitating pulp regeneration and dentin remineralization [11].
“11. Tan, J.; Xu, X.; Lin, J.; Fan, L.; Zheng, Y.; Kuang, W. Dental Stem Cell in Tooth Development and Advances of Adult Dental Stem Cell in Regenerative Therapies. Curr Stem Cell Res Ther 2015, 10, 375-383, doi:10.2174/1574888x09666141110150634.”
- lines 71-74: And it seems that inducing inflammation rapidly at the initial stage of pulp damage, controlling the severity of inflammation during the process, and inhibiting inflammation at an appropriate time to avoid adverse effects are key factors in promoting the repair of inflamed pulp [15].
“15. Cooper, P.R.; Holder, M.J.; Smith, A.J. Inflammation and regeneration in the dentin-pulp complex: a double-edged sword. J Endod 2014, 40, S46-51, doi:10.1016/j.joen.2014.01.021.”
Comments 4: Please spell out abbreviations where they first appear.
Response 4: Thanks for your suggestion. We have carefully reviewed the entire manuscript and spelled out the abbreviations as they first appear.

Reviewer 3 Report
Comments and Suggestions for Authors
This is a comprehensive review of modern dental materials used for vital pulp therapy. The topic is important and timely and the article is well written and clear. However, I have a few comments that the authors should consider:
The details on literature search and inclusion of studies are described very superficially in the abstract, while there is no description in the main text. Would it be possible to expand this section to provide more details about the study selection process (e.g. a detailed search strategy, inclusion/exclusion criteria, assessment of risk of bias, etc.)? Without these details, the potential for bias or selective reporting remains unaddressed and the reader cannot adequately assess the transparency and weight of evidence for the claims made.
Please also consider the use of the terms “bioactive” and “bioceramic”. I am aware that these terms are deeply rooted in our field (as is the term MTA, which is notoriously meaningless yet unavoidable because it is used so frequently by manufacturers, clinicians, and researchers alike), but the criticism of their potential ambiguity and susceptibility to overuse in a marketing context should be mentioned in the manuscript. In particular, there is no universally accepted standard definition of what "bioactive" material should mean since the term is often applied broadly to any material that has any interaction with tissue or even just releases ions without any biological interaction. The use of the same term for a very heterogeneous group of materials without explicit criteria for classification has rendered the term rather meaningless. The term "bioceramic" can also be misleading; better and more descriptive alternatives have been proposed, such as calcium silicate materials or hydraulic materials. Please consider addressing this terminological issue by at least mentioning that it is currently not possible to avoid these terms, even though we know they are quite inaccurate.
Would it be possible to distinguish between different levels of evidence? This is not apparent to the reader due to the lack of transparent assessment of study quality. Please also acknowledge the heterogeneity of study designs that can lead to different levels of evidence.
The manuscript could be further strengthened by synthesizing the conflicting evidence and pointing out inconsistencies in the literature. The studies that report divergent or less favorable results should also be included and discussed.
Author Response
To Reviewer #3: This is a comprehensive review of modern dental materials used for vital pulp therapy. The topic is important and timely and the article is well written and clear. However, I have a few comments that the authors should consider.
Response: We sincerely thank you for the time and energy you expended on reviewing our paper. Your comments provide valuable insights to refine its contents and analysis. We try to address the issues raised as best as possible. In the manuscript, the revised text according to the reviewer’s comments is highlighted in yellow. In the following sections, you will find our responses to your points and suggestions.
Comments 1: The details on literature search and inclusion of studies are described very superficially in the abstract, while there is no description in the main text. Would it be possible to expand this section to provide more details about the study selection process (e.g. a detailed search strategy, inclusion/exclusion criteria, assessment of risk of bias, etc.)? Without these details, the potential for bias or selective reporting remains unaddressed and the reader cannot adequately assess the transparency and weight of evidence for the claims made.
Response 1: Thank you for the careful reviewing. The inclusion/exclusion criteria and search terms have been incorporated in Section II, page 2, as shown below:
“2.Inclusion and Exclusion Criteria
The inclusion criteria were as follows:
- Original articles
- Human or animal cell culture studies, or animal studies
The exclusion criteria were:
- Case study reports
- Review or systematic review
- Commentaries/letters to the editor/expert opinion
- Non–English language articles
Search Methodology
The MEDLINE/PubMed library databases were queried for relevant articles to the topic of application of bioactive materials in VPT published up to December 2023 (last accessed December 31, 2023). The search terms were the following key words used in various combinations: “Mineral trioxide aggregates”, “Biodentine”, “iRoot BP Plus”, “pulp capping”,“lithium”, “zinc”, “Strontium”, “Magnesium”, “Silver” “inflammation”, “molecular mechanism”, “signaling pathway”, “dental stem cell”, “apical papilla stem cell”, “dental pulp”.
An initial literature search using different combinations of the search terms yielded 1112 articles (Table 1). Figure 1 presents a flowchart of the review process. Titles and abstracts of these articles were reviewed by 2 independent examiners who excluded nonqualifying publications.”
Table 1. The initial search results.
|
Search Terms |
Search Results |
|
Protein + dental pulp + inflammation |
561 |
|
Lithium/Zinc/Strontium/Magnesium /Silver + dental pulp |
61 |
|
Mineral trioxide aggregate/Biodentine/iRoot BP Plus + inflammation/ signaling pathway/molecular mechanism/vital pulp therapy |
490 |
|
Total |
1112 |
Figure 1. The initial search results.
Comments 2: Please also consider the use of the terms “bioactive” and “bioceramic”. I am aware that these terms are deeply rooted in our field (as is the term MTA, which is notoriously meaningless yet unavoidable because it is used so frequently by manufacturers, clinicians, and researchers alike), but the criticism of their potential ambiguity and susceptibility to overuse in a marketing context should be mentioned in the manuscript. In particular, there is no universally accepted standard definition of what "bioactive" material should mean since the term is often applied broadly to any material that has any interaction with tissue or even just releases ions without any biological interaction. The use of the same term for a very heterogeneous group of materials without explicit criteria for classification has rendered the term rather meaningless. The term "bioceramic" can also be misleading; better and more descriptive alternatives have been proposed, such as calcium silicate materials or hydraulic materials. Please consider addressing this terminological issue by at least mentioning that it is currently not possible to avoid these terms, even though we know they are quite inaccurate.
Response 2: Thanks a lot for the careful reviewing. We have given your suggestion careful consideration, and we believe it to be accurate. The term "bioceramic materials" has been substituted with "calcium silicate materials" wherever it appears. And the term "bioactive materials" has been specifically defined in the section where it is first used on page 2, lines 63-67.
Comments 3: Would it be possible to distinguish between different levels of evidence? This is not apparent to the reader due to the lack of transparent assessment of study quality. Please also acknowledge the heterogeneity of study designs that can lead to different levels of evidence.
Response: Thank you for your suggestion. In response to your suggestion, we have added Section III: Quality Assessment, on page 3, lines 107-121, and have differentiated the level of each document in the subsequent table, as shown below:
“3.Quality assessment
In order to enhance the transparency and reliability of the studies, a brief assessment of the quality of the included studies was conducted, and the evidence was categorized into the following three levels based on the GRADE framework: high quality, moderate quality, and low quality. Concurrently, given the heterogeneity of study designs, particular attention was devoted to the randomization of trials, the sample size, the configuration of control groups, and the objectivity of outcome assessment.
- High quality: randomized controlled trials and repeated experiments, subjects are dental pulp stem cells or other stem cells within the pulp chamber.
- Moderate quality: small sample size experiments or non-randomized controlled trials, subjects are other cells with stemness.
- Low quality: uncontrolled trials, subjects are mismatched cell types or cells of unknown origin, or animal experiments, or trials that do not fully meet the criteria for high and moderate quality.”
Furthermore, we have acknowledged the heterogeneity of the study designs in Section VII: Conclusion, page 17, lines 618-628, as shown below:
“A notable heterogeneity characterized the design of the studies incorporated within the scope of this review, encompassing in vitro cellular experiments on diverse cell types of human and animal origin, along with animal model studies employing rats or mice. These variations have the potential to influence the comparison and integration of study results. Additionally, the experimental conditions (e.g., the employed inflammation model, the concentration of materials utilized) and outcome metrics (e.g., expression levels of inflammatory factors, biocompatibility assessment) exhibited significant variation across studies. Consequently, although the present analysis offers a comprehensive understanding of the role of bioactive materials in pulpal inflammation modulation and regeneration, readers should exercise caution when directly comparing different studies.”
Comments 4: The manuscript could be further strengthened by synthesizing the conflicting evidence and pointing out inconsistencies in the literature. The studies that report divergent or less favorable results should also be included and discussed.
Response 4: Thank you for the careful reviewing. As detailed in Part VI: Inflammatory response induced by BMs in VPT(pages 10, lines 315-320 and page 11, lines 354-359), we have thoroughly analyzed and discussed the contradictions between different literatures, as shown below:
“These findings indicate that MTA induces varying inflammatory responses in different cell types. Even within the same cell type, different formulations of MTA elicit distinct inflammatory reactions. In addition, MTA may exert anti-inflammatory or pro-inflammatory effects that are subject to dynamic adjustment according to the time of action throughout the repair process.”
“Additionally, a study on MTA-induced murine healthy RAW264.7 macrophages demonstrated that cellular autophagy could be induced within 24 h, which is inconsistent with previous researches [92]. This discrepancy may be attributed to differences in autophagy regulatory mechanisms across species. However, there is currently no relevant research on inflammatory DPSCs or other human cells.”

Round 2
Reviewer 2 Report
Comments and Suggestions for Authors
The authors addressed satisfactorily all of my comments and the manuscript has been improved.
Reviewer 3 Report
Comments and Suggestions for Authors
Thank you for revising the manuscript